# Rapid soil organic carbon decomposition in river systems: effects of the aquatic microbial community and hydrodynamical disturbance

Man Zhao [1], Liesbet Jacobs [1], Steven Bouillon [1], Gerard Govers [1]

[1] KU Leuven, Department of Earth and Environmental Sciences, 3001 Leuven, Belgium

*Correspondence to*: Man Zhao (man.zhao@kuleuven.be)

**Abstract.** Different erosion processes deliver large amounts of terrestrial soil organic carbon (SOC) to rivers. Mounting evidence indicates that a significant fraction of this SOC, which displays a wide range of ages, is rapidly decomposed after entering the river system. The mechanisms explaining this rapid decomposition of previously stable SOC still remain unclear. In this study, we investigated the relative importance of two mechanisms that possibly control SOC decomposition rates in aquatic systems: (i) in

the river water SOC is exposed to the aquatic microbial community which is able to metabolise SOC much more quickly than the soil microbial community and (ii) SOC decomposition in rivers is facilitated due to the hydrodynamic disturbance of suspended sediment particles. We performed different series of short-term (168 h) incubations quantifying the rates of SOC decomposition in an aquatic system under controlled conditions. Organic carbon decomposition was measured continuously through monitoring dissolved $O_2$ (DO) concentration using a fiber-optic sensor (FirestingO$_2$, PyroScience). Under both shaking and standing conditions,

we found a significant difference in decomposition rate between SOC with aquatic microbial organisms added (SOC+AMO) and without aquatic microbial organisms (SOC-AMO). The presence of an aquatic microbial community enhanced the SOC decomposition process by 70–128 % depending on the soil type and shaking/standing conditions. While some recent studies suggested that aquatic respiration rates may have been substantially underestimated by performing measurement under stationary conditions, our results indicate that the effect of hydrodynamic disturbance is relatively minor, under the temperature conditions,

the soil type and for the suspended matter concentration range used in our experiments. We propose a simple conceptual model explaining these contrasting results.

## 1 Introduction

Rivers play an important role in the global carbon cycle by linking terrestrial and aquatic ecosystems. Each year, rivers receive and deliver large amounts of terrestrial organic carbon to the oceans (Raymond and Bauer, 2001;Ward et al., 2017). However,

rivers do not just transport OC. In 2007, Cole et. al published the idea of "rivers as an active pipe", which highlights the fact that rivers do not only transport but also process large amounts of OC (Cole et al., 2007). Mounting evidence indicates that a significant fraction of the OC processed in rivers is soil organic carbon (SOC), including dissolved (DOC) and particulate (POC) forms. The mobilized SOC can display a very wide range of ages among rivers depending on the carbon sources and retention time. Many studies have demonstrated that the DOC transported by large rivers such as the Amazon and Mississippi is generally quite young

(Mayorga et al., 2005, Rosenheim et al., 2013), while in other systems the POC is often associated with relatively old radiocarbon ages (in the range of 1000–5000 y B.P.) (Marwick et al., 2015; Raymond and Bauer, 2001; Dodds and Cole, 2007; Mccallister and Del Giorgio, 2012). Several studies have shown that, when SOC is delivered to a river, part of this SOC can be rapidly mineralized and emitted back to the atmosphere (Lapierre et al., 2013; Wilkinson et al., 2013; Mayorga et al., 2005). Understanding the mechanisms that contribute to this active mineralization process of SOC is essential for understanding the role of rivers in the

global carbon cycle and for assessing how the metabolism of rivers may respond to environmental perturbations such as an increase

or decrease of terrestrial carbon delivery to the river system and/or changes in hydrology and climate. Indeed, the amount of SOC imported into riverine ecosystems is often large compared to the autochthonous within-river primary production (Cole and Caraco, 2001). It is therefore important to understand the fate of this terrestrial SOC as this does not only affect the global carbon cycle but also strongly regulates the ecological functioning of river ecosystems (Kling, 1995). A key question is then which factors control the decomposition of terrestrial SOC, which has been stable in soil for decades to centuries, when it enters into a river system.

In recent years, the mechanisms controlling this rapid riverine mineralization process have gained increasing attention (Aufdenkampe et al. 2011; Guenet et al. 2014; Ward et al. 2017). While in transit, SOC can be degraded by microbial mineralization (Ward et al., 2013) and photochemical oxidation (Spencer et al., 2009). As can be expected, these processes are closely associated with a suite of factors such as temperature (Lapierre et al., 2013; Gudasz et al., 2015), the availability of oxygen (Koehler et al., 2012), and the presence and composition of microbial communities (Ward et al., 2019), along with physical river properties, such as river velocity and hydrodynamic conditions (Ward et al., 2018). For example, Ward et al. (2018) incubated river water and sediment under three rotation regimes to mimic river flow and found that respiration rates in rotating samples were 1.4 (under 0.22 $m·s^{-1}$) and 2.4 times (under 0.66 $m·s^{-1}$) higher compared with stationary conditions. Mechanical disturbance can be expected to enhance mineralization because the physical breakup of large particles induced by river water disturbance may increase the accessibility of microbial enzymes to SOC (Lal, 2003; Richardson et al., 2013).

Microbial organisms influence SOC cycling not only via decomposition but also because microbial products are themselves important components of soil organic matter (Kögel-Knabner, 2002). The interactions of microbial communities and SOC decomposition have been extensively studied in terrestrial ecosystems (Cleveland et al., 2007; Hu et al., 2014; Tian et al., 2016). The interactions between microbes and SOC in aquatic systems received until recently little attention. Recently, Ward et al. (2019) illustrated the potential importance of the composition of the aquatic microbial community by showing that mixing the water of two lowland tributaries (the Tapajós and Xingu rivers) and the Amazon main river resulted in enhanced respiration rates, which they attributed to the fact that such mixing resulted in a more diverse microbial community capable of digesting more OC. While these recent studies clearly indicate that different mechanisms do indeed control OC processing in rivers, important knowledge gaps do remain with respect to the specific controls of SOC decomposition by aquatic microbes. One of the reasons for the latter is that, hitherto, very few experiments were carried out with a factorial design allowing to investigate the relative importance and potential interaction of different individual factors.

The main objective of this study is to shed light on the fate of SOC in river systems by investigating the relative importance of two key mechanisms that were previously suggested to potentially contribute to the rapid decomposition of previously stable SOC in aquatic systems: (i) in the river water, SOC is exposed to an aquatic microbial community which may be able to metabolise SOC much more quickly than the soil microbial community and (ii) SOC decomposition in rivers may be facilitated due to the hydrodynamic disturbance of sediment. We base our analysis on continuous measurements of DO consumption during a series of lab incubation experiments and the characterisation of the evolution of SOC, POC and DOC characteristics throughout the incubation period. By doing so we were able to quantify the importance of these mechanisms both in absolute and relative terms.

## 2 Material and methods

### 2.1 Site description

We sampled the Dijle river located ca. 2 km upstream of the city of Leuven, which is situated in the central Belgian loess belt (Fig. 1). The relief of the 700 $km^2$ large Dijle river catchment upstream of Leuven varies from about 25 m above sea level (a.s.l) in the north to ca.165 m a.s.l in the south. The catchment is characterized by an undulating plateau in which several brooks and rivers are

incised. Slope gradients are usually less than 5 %, although maximum slopes of ca. 50% can be found along the valleys. The majority of the soils are Luvisols, developed in the loess deposits (FAO, 1998). The land use in the catchment is mainly cropland, which is particularly vulnerable to erosion during spring and early summer when vegetation cover is low and rainfall intensity is high (Takken et al., 1999). The contemporary land use in the floodplain is dominated by grassland, plantation forests and built-up area (Broothaerts et al., 2014).

## 2.2 Experimental approach

**Experimental setup**

Our measurement approach involved collecting a large volume of river water, subsequently treated by removing all suspended particles via filtration and dispensing the filtrate into a series of 320 ml glass incubation bottles. The treatments included (1) SOC with rotation, (2) SOC under stationary condition, (3) DOC control with rotation, (4) DOC control under stationary condition, (5) SOC with rotation without AMO, and (6) SOC under stationary condition without AMO (see Table 1 for details). Mechanical breakup was simulated by a custom-designed swing system (Fig. 2), which kept the particles in suspension during the incubation. Incubation bottles were fixed in the PVC clip brackets, and rotated at about 7 rounds per minute. Since the bottles were connected with the FirestingO$_2$ fibre, the swing system was first rotated 180° anticlockwise and then 180° clockwise back, which together was counted as one round. In order to get a higher disturbance intensity, 8 g of glass beads (diameter: 2 mm) were added to the bottle before adding soil and water samples. The glass beads were pre-sterilized at 450 °C for 1 hour to avoid contamination.

**Collection and incubation**

River water was collected from the Dijle river (details in Table 2). The water was filtered on 0.7 µm glass fiber filters to remove all suspended particles. For treatments without aquatic microbial communities, river water was then filtered through 0.2 µm syringe filters. Since filtration at 0.7 µm already removed the majority of microbial organisms, 4 ml unfiltered river water (ratio of inoculum to the incubated water sample: 1:79) was added to serve as an inoculum for treatments with aquatic microbial organisms. Soil samples were collected from a Belgian Loess Belt near Leuven: one from arable land (50°48'31.6″N, 4°35'16.9″E), which was collected from a depth of 0-20 cm; the other from Bertem forest (50°52'59.8″N, 4°38'24.2″E). The top litter layer was first removed, then soil sample was collected from a depth of 0-20 cm. These soils were oven dried at ~55 °C and then sieved through 2 mm to remove all roots and stones. In order to obtain a detectable rate of oxygen consumption, the POC concentration was controlled at 10–12 mg L$^{-1}$ by adding 160 mg arable soil and 60 mg forest soil in 320 ml river water (details in Table 3). While the sediment and POC concentrations we used in our experiments are relatively high, they are not unrealistic: during high flood we observed POC concentrations exceeding 10 mg L$^{-1}$ in ca. 5% of our samples. The bottles were subsequently closed without headspace. For the suspended condition treatments, bottles were fixed on the swing systems (Fig. 2). The remaining bottles were placed on a shelf to keep particles settled in each bottle. To avoid the influence of temperature and light (both potentially influencing OC mineralization), all bottles were wrapped with aluminium foil and incubated in a temperature-controlled room at ~20 °C. For each experiment, 4 incubation bottles with the same treatment were incubated, which allowed us to sample for POC and DOC during the course of the experiment (at t = 24 h, 48 h, 96 h, 168 h). Six experimental runs for each soil type were conducted to investigate the effect of water disturbance and the presence of an aquatic microbial community on SOC decomposition.

Given that the soil samples were oven-dried at ~55 °C before the incubation, a certain amount of soil microbial organisms might have been eliminated. In order to test this, we carried out a series of supplementary experiments whereby air dried and oven dried soil material was used under stationary and rotation conditions. Soil material was collected from the same arable land (OC%:

1.5 %). The soil was divided into two parts and dried with two methods: one part was oven dried at ~55 °C, the other was air dried. The incubations followed the same procedures mentioned above.

**DO, POC and DOC measurements**

DO was measured every 10 s for 168 hours using an optical oxygen meter (FireSting$O_2$), and we used the DO data to derive the total amount of C mineralized in each incubation series, using a procedure similar to the one used in previous studies (Berggren et al., 2012; Richardson et al., 2013). It is known that, when OC is decomposed, the $O_2$: $CO_2$ ratio is not constant but depends on the composition of the substrate and the characteristics of the bacterial community (Berggren et al., 2012). Richardson et al. (2013) took the elemental composition of organic matter into account, and reported an $O_2$: $CO_2$ ratio of 1.04–1.2 for detrital organic matter and 1.0 for DOC. In this study, we used an $O_2$: $CO_2$ ratio equal to 1.0 for both SOC and DOC, thus assuming that the mineralization of 1 mole C consumed 1 mol $O_2$. Using a respiration ratio of 1.0 might result in an overestimate of the total amount of C mineralized, but since all treatments were calculated with the same respiration ratio and the same soil samples were used throughout the experiments, the relative variations will not be affected by this choice.

After the incubation period, the entire volume of ~320 ml water was filtered for later determination of total POC, particulate nitrogen (PN) and the stable C isotope ratios ($\delta^{13}C$) of POC on pre-combusted 25 mm Whatman GF/F filters (pore size: 0.7 μm). Due to the high concentrations of soil material in each bottle, each water sample was filtered on 2–5 filter papers. For arable soil incubations, water samples were filtered on 5 filter papers, we measured all filter papers for 2 repetitions, and 2 of 5 filters papers for the other 4 repetitions. Then we used the suspended sediment weight and POC concentrations measured on this subsample to calculate the POC and PN content as well as the $\delta^{13}C$ signature of the entire sample. For forest soil incubations, water samples were filtered on 2–5 filter papers. All of the filter papers were oven dried at 50 °C and preserved for POC, PN and $\delta^{13}C$ measurements. Inorganic C was removed from the filters by exposing them to HCl fumes overnight in a desiccator. Subsequently, the dried filters were packed in Ag cups for analysis on an elemental analyser-isotope ratio mass spectrometer (EA-IRMS, ThermoFinnigan Flash HT and Delta V Advantage). Certified (IAEA-600, caffeine) and in-house laboratory standards (leucine and tuna tissue) were analysed throughout each run.

To determine the DOC concentration and its stable isotope composition, 40 ml water samples (filtered at 0.2 μm) were collected and stored in glass vials with Teflon-coated screw caps and 100 μL of $H_3PO_4$ was added for preservation. Analysis of DOC and $\delta^{13}C_{DOC}$ was performed on a wet oxidation TOC analyzer (IO Analytical Aurora 1030W) coupled with an isotope ratio mass spectrometer (ThermoFinnigan Delta V Advantage). Quantification and calibration were performed with IAEA-C6 ($\delta^{13}C =$ − 10.4 ‰) and an internal sucrose standard ($\delta^{13}C = -26.99 \pm 0.04$ ‰).

The two soil samples used were analyzed for their $^{14}C$ content on a MICADAS accelerator mass spectrometry system at the Royal Institute for Cultural Heritage (KIK/IRPA, Brussels). Data are reported in years BP, whereby 1950 is taken as the present.

**2.3 Statistical analysis**

Statistical tests were performed in R (version 3.5.1, R Core team). The normality of data was tested with the Shapiro–Wilk test. The paired sample t test was used to test for differences in DO consumption rates, total amount of C mineralized and POC loss between treatments under rotation and stationary conditions as well as to test for differences between treatments with and without the presence of AMO. To identify the combined effect of AMO and physical disturbance on the C decomposition rates, we used two-way ANOVA with the presence of AMO and mechanical rotation as the main factors. ANOVA was carried out separately for the two soil types. Average values are given ± the standard deviation.

# 3 Results

## 3.1 Effect of water disturbance and AMO on DO consumption rates

The initial DO concentration in the river water varied from 7.58 to 10.67 mg L$^{-1}$. In all treatments, DO concentrations followed a decreasing trend, and the DO consumption rates were relatively constant over time (Fig. 3). For incubations where soil was present, cumulative DO consumption ranged between 0.7 and 2.6 mg L$^{-1}$ during the 168-h incubation period. The highest DO consumption occurred in the SOC+AMO treatments with an average consumption rate of $0.015 \pm 0.005$ mg O$_2$ L$^{-1}$ h$^{-1}$ (arable soil) and $0.010 \pm 0.002$ mg O$_2$ L$^{-1}$ h$^{-1}$ (forest soil). DO consumption was lowest for treatments where soil was present without AMO with an average

of $0.007 \pm 0.002$ mg O$_2$ L$^{-1}$ h$^{-1}$ (arable soil) and $0.004 \pm 0.001$ mg O$_2$ L$^{-1}$ h$^{-1}$ (forest soil). We found that keeping soil particles in suspension resulted in a relatively small acceleration of the OC decomposition process. With the presence of aquatic microbial organisms (SOC+AMO), DO consumption rate was increased on average by 13 % ($p < 0.05$) for the arable soil while no significant effect was found for forest soil ($p > 0.05$) (Table 4).

The addition of AMO, on the other hand, had a much stronger stimulation effect on OC decomposition. Compared with SOC-

160 AMO treatments, DO consumption rates were approximately doubled by the presence of AMO for both soil types. This increase was present under both rotation (arable soil: $0.015 \pm 0.005$ vs. $0.007 \pm 0.002$ mg O$_2$ L$^{-1}$ h$^{-1}$, $p < 0.01$; forest soil: $0.010 \pm 0.002$ vs. $0.005 \pm 0.002$ mg O$_2$ L$^{-1}$ h$^{-1}$, $p < 0.01$) and stationary conditions (arable soil: $0.014 \pm 0.003$ vs. $0.007 \pm 0.002$ mg O$_2$ L$^{-1}$ h$^{-1}$, $p < 0.01$; forest soil: $0.009 \pm 0.002$ vs. $0.004 \pm 0.001$ mg O$_2$ L$^{-1}$ h$^{-1}$, $p < 0.01$) (Table 4). In addition, the presence of AMO and rotation had no significant combined effect on the C decomposition rates for both soil types ($p > 0.05$).

## 3.2 Total amount of C mineralized

We calculated the total amount of C mineralized in each treatment, using an O$_2$: CO$_2$ ratio of 1.0 and expressed mineralisation rates on a per carbon basis. Obviously, trends in total C mineralisation are similar to those in oxygen consumption. For SOC+AMO incubations with arable soil, keeping particles in suspension increased the total amount of C mineralized by 13 % ($p < 0.05$, Fig. 4). Also for forest soil a small increase was noted (11 %) but this was not statistically significant ($p > 0.05$). Compared with SOC-

170 AMO, the presence of aquatic microbial organisms (SOC+AMO) – regardless of soil type and rotating/stationary conditions – significantly stimulated OC decomposition (Fig. 4). For arable soil, AMO addition led to 91 % (under rotation conditions, $p < 0.01$)–128 % (under stationary conditions, $p < 0.01$) more C mineralized by the end of the incubation, and 116 % (under rotation conditions, $p < 0.01$)–118 % (under stationary conditions, $p < 0.01$) with forest soil depending on rotation and stationary conditions. In addition, when comparing the mineralisation of oven-dried and air-dried soil without AMO addition, oven-dried soil incubation

showed a 40 % lower C loss in comparison to air dried soil incubation at the end of the incubation (Fig. 5, $p < 0.05$). However, the addition of AMO resulted in a significantly higher C loss with both treatments (air drying: 70 %, $p < 0.05$; oven drying:165 %, $p < 0.05$).

## 3.3 Particulate and dissolved organic carbon concentrations and $\delta^{13}$C$_{POC}$, $\delta^{13}$C$_{DOC}$ values

Measurements of final POC concentrations showed a reduction of POC at the end of the incubations, where 5–13 % of POC was

180 mineralized for incubations with arable soil, and 1–11 % for incubations with forest soil (Table 4). Keeping particles in suspension resulted in ca. 4 % more POC loss in SOC+AMO incubation series for both soil types, although the increase was only significant for arable soil (arable soil: 13 % vs. 9 %, $p < 0.05$; forest soil: 11 % vs. 7 %, $p > 0.05$, Table 4). Conversely, keeping particles in suspension in SOC-AMO incubation series showed negative effects with 1 (arable soil) to 4 % (forest soil) less POC loss, but this difference was not statistically significant ($p > 0.05$). For both soils, the presence of AMO led to ca. 8–10 % more POC loss under

rotation conditions ($p < 0.05$). This effect was smaller (2–3 %) and non-significant ($p > 0.05$) under stationary conditions (Table 4).

These trends are generally consistent with the patterns derived from the oxygen consumption measurements, i.e. a larger reduction in POC when AMO is present and a limited effect of hydrodynamic disturbance on POC decomposition. However, it is also clear that variations in residual POC are less consistent than those observed from oxygen consumption rates. In 7 of 12 incubation series, the combined POC and DOC losses exceeded the total amount of C mineralized calculated from DO consumption (Table 4).

Variations in $\delta^{13}C_{POC}$ values showed a similar pattern for both soil types (Fig. 6): firstly, an increase in $\delta^{13}C_{POC}$ occurred during the first 24 h (for forest soil) or 48 h (arable soil). The increase in $\delta^{13}C_{POC}$ appears to be more important for forest soil (0.4 ‰– 0.6 ‰) in comparison to arable soil (0.2 ‰–0.4 ‰). After this initial period, $\delta^{13}C_{POC}$ values stabilised.

DOC concentrations were relatively stable during the incubation (Fig. 7; Table 4). Initial $\delta^{13}C_{DOC}$ value was lower in arable soil experiments where no AMO were present. When AMO were present the initially higher $\delta^{13}C_{DOC}$ values declined to similar values as those observed in experiments without AMO within the first 48 h. In the experiments with forest soil initial $\delta^{13}C_{DOC}$ values were slightly but not significantly higher (0.6 ‰, $p > 0.05$) than the initial $\delta^{13}C_{DOC}$ values for arable soil without AMO. $\delta^{13}C_{DOC}$ values were stable throughout the incubation period for all forest soil experiments.

For both soil types, the equilibrium $\delta^{13}C_{DOC}$ values appear to be somewhat lower than the equilibrium $\delta^{13}C_{POC}$ values: this difference was more pronounced for the forest soil (ca. 1 ‰ in comparison to ca. 0.5 ‰ on average), but non-significant in both cases ($p > 0.05$).

## 4 Discussion

Our results show that a fraction of the terrestrial SOC can indeed be mineralized relatively quickly when introduced in an aquatic environment. With the presence of aquatic microbial organisms (SOC+AMO), up to 0.58 (forest soil) or 0.97 (arable soil) mg C $L^{-1}$ was mineralized within the 168 h incubation period, equivalent to 83–139 µg C $L^{-1}$ $d^{-1}$ or 9-12 µg C $(mg\ SOC)^{-1}d^{-1}$. Comparable respiration rates of 20–80 µg C $L^{-1}$ $d^{-1}$ were reported under similar temperature conditions by Berggren et al. (2010). Similar incubation experiments with water samples collected from northern temperate lakes and streams reported respiration rates from 16 to 54 µg C $L^{-1}$ $d^{-1}$ (McCallister and Del Giorgio, 2012). We found that C mineralization rates were quite significant: in the presence of an AMO (which is always the case in natural conditions), 4 to 6 % of the OC present at the initiation of the experiment was mineralized during the experiment (Fig. 4). Typical soil incubation experiments show a loss of max. 2 % of the SOC in the first two weeks: Gillabel et al., (2010) incubated soil samples under 25 °C and found ca. 2 % SOC was respired in 20 days with a mineralization rate at around 1 µg C per mg C per day; Angst et al., (2019) conducted similar soil incubation and found a cumulative respiration of 20 mg $CO_2$ (g C soil)$^{-1}$ in 14 days, equivalent to around 0.4 µg C per mg C per day. Li et al., (2018) incubated soil with fertilization and straw application, and found a highest cumulative respiration at around 800 mg $CO_2$-C $kg^{-1}$ soil in 20 days, equivalent to around 2 µg C per mg C per day. Thus mineralization rates observed in soil incubation experiment (0.4–2 µg C per mg C per day) are much lower than those observed in our study, where 6–9 µg C is mineralized per mg C per day. This suggests that SOC indeed decomposes 5-10 times more rapidly in aquatic systems than in the terrestrial environment and that SOC is not as recalcitrant as preciously thought in aquatic systems (Mayorga et al., 2005; McCallister and Del Giorgio, 2012). Several studies already suggested that the transition from terrestrial to aquatic conditions likely facilitated SOC decomposition rates because of shifts in environmental conditions (Butman and Raymond, 2011; McCallister and Del Giorgio, 2012). In soils, sorption of OC to mineral surfaces and encapsulation of C within soil aggregates may protect SOC from complete mineralization (Bianchi et al., 2011; Schmidt, et al., 2011). This results in the accumulation of older SOC in pools that are less accessible to

decomposers and their extracellular enzymes (Marín-Spiotta et al., 2014). When SOC enters aquatic systems, a disruption of the mechanisms protecting C from mineralization, such as a physical disturbance due to the physical action of transport in water but also due to aggregate slaking (Le Bissonnais, 1996), may lead to the exposure of these protected pools to decomposers and therefore to an increase of the SOC decomposition rate. Alternatively, SOC decomposition may be accelerated due to the fact that, in an aquatic environment, a population of possible consumers is present that is different from that in the soil and that may be capable of rapidly mineralizing SOC that is otherwise preserved over long-time scales in a soil environment (Wu et al., 2018; Ward et al., 2019b).

Our results indicate that physical soil disturbance, which was simulated by rotation in our experiments, had a relatively minor effect on SOC decomposition (arable soil: $p < 0.05$; forest soil: $p > 0.05$) (Fig. 3, Fig. 4). This result is different from that reported in two previous studies with similar incubation approaches. Richardson et al (2013) and Ward et al (2018) found that keeping sediment particles in suspension increased river-borne organic matter decomposition by 40 to 140 %, depending on the rotation speed. The differences in experimental settings, such as POC composition, POC concentration and microbial community may influence the POC decomposition rates (Weyhenmeyer et al., 2012; Ward et al., 2019a). However, these differences as such do not explain the minor effect of physical disturbance observed in this study. Thus, our data do indicate that the relative role of physical disturbance vs. that of exposure to an aquatic microbial community may vary considerably between different ecosystems. We propose two possible reasons as to why we did not find a strong effect of physical disturbance on SOC decomposition. It is hypothetically possible that the level of physical protection of (a large part of) the SOC that is present is such that it is not disrupted by the physical disturbance that we imposed. Alternatively, the level of physical protection was so weak that the mere immersion of soil particles in water was sufficient to destroy most of it. Given the fact that loess soils are known for their very low structural stability (Le Bissonnais, 1996) and the high POC mineralisation rates we observed, we propose that the latter is more likely in our experiments. The results of our last series of experiments (Fig. 5) also show that, when SOC is introduced into an aquatic environment where no aquatic microbial community is present, there is also a significant degree of SOC decomposition. Again, this suggests that the simple immersion into water results in the breakdown of the physical protection of SOC, so that a similar microbial community becomes much more effective in decomposing SOC in an aqueous rather than a soil environment. We found that mineralization rates of forest SOC were lower than those of arable land SOC (Fig. 4). Forest soils often have a relatively stronger structural stability associated with a higher SOC content (Göl, 2009, Gajic et al., 2006). If the lower mineralization rates of forest SOC would indeed be due to a higher structural stability of the forest soil aggregates, one would therefore expect that mechanical disturbance would have a stronger effect on SOC mineralization for the forest soil. However, the effect of mechanical disturbance is small and statistically insignificant, also for the forest soil. Therefore, it seems more likely that the difference in response we observed between the two SOC types is due to differences in the composition of the organic matter in both soils. The forest SOC has a much higher C/N ratio, which is generally associated with a lower decomposability (Liang et al., 2017).

The presence of an aquatic microbial community caused a much more rapid mineralization of SOC (Fig. 3, Fig. 4). In all experimental runs that we performed (n = 24), there was a significant difference between treatments with and without aquatic microbial organisms ($p < 0.01$). Given that the soil samples were oven-dried at ~55 °C before the incubation, the effect of inoculation with AMO may at least partly be explained by the fact that the soil microbial community was killed by the drying process. Comparing the results of oven-dried and air-dried soil without AMO addition, it is clear that oven drying indeed led to a 40 % decrease in the total amount of C mineralized in comparison to air dried soil, indicating that oven drying indeed eliminated an important fraction of OC-consuming soil microbial organisms (Fig. 5). However, the addition of AMO resulted in clearly higher C losses with both soil treatments (air drying: ca.70 %; oven drying: ca.165 %), indicating that aquatic microorganisms indeed have the capacity to rapidly consume SOC that is not readily mineralized by soil microorganisms. It is well-known that a significant

fraction of the SOC is relatively old (Trumbore, 2000). The presence of such old fractions in the soil suggest that the soil ecosystem does contain no or only a very small number of consumers capable of mineralizing this POC fraction. Some consumers in aquatic ecosystems may have the capability to consume this relatively old POC relatively quickly, especially when it is no longer physically protected.

The higher POC consumption rates observed when an AMO is present may, of course, partly be due to the fact that more microbes were present in those experiments where AMO were introduced. We compared the initial population of bacteria with and without the addition of AMO. The addition of AMO led to 20–30 % more bacteria present in the water at the beginning of the experiment (arable soil: $4.30 \times 10^5$ $vs.$ $3.52 \times 10^5$ cells ml$^{-1}$; forest soil: $3.67 \times 10^5$ $vs.$ $2.86 \times 10^5$ cells ml$^{-1}$). This larger initial population could partly explain the higher SOC decomposition rates with the addition of AMO, but the fact that the addition of AMO increases SOC decomposition rates by 70–165 % rather than 20–30 % does suggest that the aquatic microbial community is indeed capable of attacking old, stable SOC more effectively than the soil microbial community. Although the microbial community is considered to play a central role in shaping OC reactivity in both terrestrial and aquatic systems (Schmidt et al., 2011), such strong stimulation effect of the addition of AMO on SOC has rarely been reported.

The evolution of POC and DOC concentrations during the experiments is generally in agreement with the patterns derived from the oxygen consumption measurements. However, the variations in residual POC are less consistent than those observed from oxygen consumption. Several reasons might explain the discrepancy between the total C mineralisation as calculated from oxygen consumption in comparison to direct measurements of POC. Firstly, POC and DOC samples were collected from 4 incubation bottles for one treatment. Even though we controlled the initial conditions of each bottle as closely as possible, there might be heterogeneity between different bottles with respect to the OC content of the soil sample that was placed into the bottle. Secondly, we compared POC measurements from 2–5 filter papers for arable soil and 5 filter papers for forest soil. For each individual measurement the filter weight has to be subtracted from the gross weight of the filter plus the sediment. Given the small quantities of sediment present on the filters, small weighing errors will result in relatively large errors in the calculation of the amount of POC that is remaining. DO measurements are non-intrusive and are not subject to measurement errors related to the weighing of small quantities. We therefore believe that the oxygen consumption measurements provide us with more robust measurements of OC decomposition in comparison to direct measurements of the OC content of the remaining sample. The direct measurements suggest that, overall, POC mineralisation was more important than DOC mineralisation when POC was present. Indeed, DOC concentrations showed little variation during the experiments, despite significant oxygen consumption rates.

The increase of $\delta^{13}C_{POC}$ values during the first 24–48 hours suggests that during this period an isotopically lighter POC fraction was preferentially mineralised. This resulted in the POC in the aquatic environment becoming enriched in $^{13}C$ by 0.2–0.6 ‰ compared to the POC in the original soil sample. After the initial adjustment period, $\delta^{13}C_{POC}$ remained stable, suggesting the initial adjustment is indeed due to the preferential consumption of a somewhat lighter, less recalcitrant POC-fraction rather than continued preferential consumption of lighter POC. The fact that, over the whole course of the experiments, the $\delta^{13}C_{DOC}$ values are lower than the corresponding $\delta^{13}C_{POC}$ values, is on the other hand, best explained by a continuous leaching/release of DOC with a somewhat lower $\delta^{13}C$ signature from the soil POC, replacing the mineralised DOC as it is unlikely that the DOC fraction would be entirely stable while POC is continuously mineralised. Mineralisation of the original DOC and its replacement with soil-derived DOC could also explain the drop in $\delta^{13}C_{DOC}$ during the initial phase of the experiments with arable soil and AMO, because the lighter DOC that was originally present in the river water is replaced by soil-derived DOC. However, this drop was not observed in the experiments with forest soil. This may be partly explained by the higher $\delta^{13}C$ value of the forest soil (–28.6 ‰) in comparison to the $\delta^{13}C$ signature of the arable soil (–29.4 ‰) causing the DOC released from the forest soil to have an isotopic signature close to that of the river water. While the patterns described above would be consistent with the preferential decomposition of isotopically

lighter POC, we did not observe an increase in $\delta^{13}C_{POC}$ during our experiments as might be expected by the selective mobilisation of an isotopically lighter soil fraction: this can be explained by the relatively small differences in $\delta^{13}C$ values between POC and DOC in combination with the fact that only a small fraction of the POC is ultimately mineralised, whereby most of this mineralised fraction may have been directly transformed to $CO_2$. If this mineralization does not selectively affect specific fractions of the POC pool, the $\delta^{13}C_{POC}$ values can be expected to remain more or less constant throughout the incubation period.

Based on our findings, we propose that the relative importance of physical disturbance vs. exposure to a novel microbial community is likely to depend on (i) the level to which the SOC is indeed physically protected and (ii) the extent to which this protection is destroyed by aggregate disruption when soil particles are introduced in river water. When the protection level is relatively important but, at the same time, sensitive to water immersion, further physical disturbance due to transport of soil particles in the river is unlikely to strongly increase SOC breakdown because aggregates are already destroyed when the sediment enters the water. If, on the other hand, physical protection is strong and aggregates are resistant to immersion, physical disturbance may be necessary to break down soil aggregates to the extent that is needed to expose a significant fraction of SOC to the microbial community present in water (Fig. 8). The loess-derived soils we used in our experiments do have a weak structure (Govers, 1991) resulting in rapid aggregate breakdown upon immersion in water and a relatively small effect of mechanical disturbance on mineralisation rates. This also explains the insignificant interaction effect of hydrodynamic disturbance and the presence of an aquatic microbial community on the C decomposition rates for both soil types ($p > 0.05$). The mere immersion of soil particles in water is sufficient to destroy most of the soil aggregates. Therefore, further disturbance would not strongly increase the interactions between soil particles and microbial organisms. The effect of the presence of an aquatic microbial community, on the other hand, will depend on its composition and its vigour. The addition of SOC may shift aquatic microbial metabolisms and make it more prone to SOC decomposition (Lennon and Pfaff, 2005). Other factors such as water temperatures and nutrient availability may also play a role. The river water we used in our study has a high nutrient content (Table 2): the availability of a large nutrient pool may allow AMO to use the SOC more effectively and may be one of the factors explaining why we saw such an important increase in SOC decomposition when AMO were added. However, it is fair to state that our current understanding of the interaction between old SOC and different microbial communities is too limited to develop general principles describing which factors may stimulate or slow the decomposition of SOC exposed to a new microbial community.

**5 Conclusions**

We investigated the relative importance of physical disturbance vs. exposure to a novel microbial community on SOC decomposition rates in aquatic environments. While some recent studies found that the impact of mechanical disturbance on SOC decomposition rates was very important, we found only a very modest increase in SOC decomposition when soil particles were mechanically disturbed and kept in suspension. A simple conceptual model, whereby the effect of mechanical disturbance is assumed to depend on the initial structural stability of soil aggregates can explain this difference in findings: mechanical disturbance is only important when soil aggregates are strong enough to withstand the disruptive forces imposed by immersion in water.

Our study also highlights the role of aquatic microbial organisms in SOC decomposition in river systems. Aquatic microbial organisms are capable of attacking SOC, leading to rapid SOC decomposition in river systems. Given the variability of aquatic microbial community composition in different aquatic systems, understanding the linkage between aquatic microbial community composition and abundance on the one hand and the resultant SOC mineralization rates on the other hand is important not only to

better understand $CO_2$ outgassing from aquatic systems, but also to understand how the ecological functioning of rivers and lakes are affected by the large changes in SOC inputs that result from human activities.

**Data availability**

All data used and produced through this study are available with the electronic supplement.

**Author contribution**

GG, SB, LJ and MZ involved in the design of the experiments, and MZ carried out the experiments. SB and MZ conducted the laboratory analysis. All authors offered advice on the data analysis and contributed to the paper preparation.

**Declaration of Competing Interest**

The authors declared that there is no conflict of interest.

**Acknowledgements**

We acknowledge the China Scholarship Council (grant no. 201706300031) for supporting Man Zhao's research at KU Leuven. We are grateful to Dr. Zita Kelemen for help with stable isotope analysis, to Dr. Filip Meysman (UA) for his suggestions with respect to experimental design and to M. Boudin (KIK/IRPA) for radiocarbon measurements. We thank the two anonymous reviewers, whose constructive comments and suggestions greatly improved an earlier version of this manuscript.

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

**Table 1.** Experimental setup for incubations under controlled laboratory conditions

| Treatment number | Treatments | Number of repetitions | Ingredients |
|---|---|---|---|
| 1 | SOC, rotation | 6 | soil + 0.7 μm filtered river water+ inoculum |
| 2 | SOC, stationary | 6 | soil + 0.7 μm filtered river water+ inoculum |
| 3 | DOC control, rotation | 6 | 0.7 μm filtered river water+ inoculum |
| 4 | DOC control, stationary | 6 | 0.7 μm filtered river water+ inoculum |
| 5 | SOC without AMO, rotation | 6 | soil + 0.2 μm filtered river water |
| 6 | SOC without AMO, stationary | 6 | soil + 0.2 μm filtered river water |

**Table 2.** Nutrient concentrations in the Dijle river water (data from the Flanders Environment Agency: https://www.vmm.be/data/waterkwaliteit)

| Year | Number of measurements | $NH_4^+$ (μmol $L^{-1}$) | $NO_3^-$ (μmol $L^{-1}$) | $PO_4^{3-}$ (μmol $L^{-1}$) |
|------|------------------------|--------------------------|--------------------------|-----------------------------|
| 2020 | 3 | $24 \pm 7$ | $717 \pm 55$ | no data |
| 2016 | 6 | $48 \pm 30$ | $410 \pm 15$ | $6 \pm 1$ |
| 2014 | 6 | $26 \pm 11$ | $537 \pm 94$ | $6 \pm 1$ |

**Note**: values are Mean ± SD.

**Table 3.** Characteristics of the two soils used in this study. Texture was determined using laser diffraction (Coulter LS 13 320).

| Soil samples | Soil texture | | OC content (%) | C/N (weight/weight) | $\delta^{13}C$ (‰) | $^{14}C$ age (yr BP) |
|---|---|---|---|---|---|---|
| Arable soil | Sand % | 23 | 2.40 | 9.9 | –29.4 | 267 ± 21 |
| | Loam % | 69 | | | | |
| | Clay % | 8 | | | | |
| Forest soil | Sand % | 39 | 5.20 | 17.6 | –28.6 | 334 ± 22 |
| | Loam % | 56 | | | | |
| | Clay % | 5 | | | | |


| | | OC content (%) | C/N (weight/weight) | $\delta^{13}C$ (‰) | $^{14}C$ age (yr BP) |
|---|---|---|---|---|---|
| Soil samples | Soil texture | | | | |

Sand %  23

**Table 4.** Total amount of C mineralized and POC, DOC loss at the end of the incubation series calculated as the difference in the mass of carbon that was introduced and the mass of carbon recovered from the final sample.

| Soil type | Treatments | Total amount of C mineralized (derived from DO, mg) | Significant (p<0.05) difference with stationary counterpart | Significant (p<0.01) difference with -AMO counterpart | POC loss (mg) | Percentage loss of POC (%) | Significant (p<0.05) difference with stationary counterpart | Significant (p<0.05) difference with -AMO counterpart | DOC loss (mg) | POC+DOC loss (mg) |
|---|---|---|---|---|---|---|---|---|---|---|
| Arable soil | SOC+AMO/rotation | $0.31 \pm 0.09$ | **Y** | **Y** | 0.5 | $13 \pm 5$ | **Y** | **Y** | 0.04 | 0.54 |
| | SOC+AMO/stationary | $0.27 \pm 0.06$ | | **Y** | 0.33 | $9 \pm 3$ | | N | 0.1 | 0.43 |
| | DOC control/rotation | $0.15 \pm 0.07$ | N | | \ | \ | | | 0.14 | \ |
| | DOC control/stationary | $0.16 \pm 0.05$ | | | \ | \ | | | 0.24 | \ |
| | SOC-AMO/rotation | $0.13 \pm 0.04$ | N | | 0.22 | $5 \pm 3$ | N | | −0.01 | 0.21 |
| | SOC-AMO/stationary | $0.14 \pm 0.04$ | | | 0.24 | $6 \pm 3$ | | | −0.01 | 0.23 |
| Forest soil | SOC+AMO/rotation | $0.20 \pm 0.03$ | N | **Y** | 0.34 | $11 \pm 7$ | N | **Y** | −0.10 | 0.24 |
| | SOC+AMO/stationary | $0.18 \pm 0.04$ | | **Y** | 0.22 | $7 \pm 5$ | | N | −0.04 | 0.18 |
| | DOC control/rotation | $0.11 \pm 0.04$ | N | | \ | \ | | | 0.12 | 0.12 |
| | DOC control/stationary | $0.13 \pm 0.08$ | | | \ | \ | | | 0.11 | 0.11 |
| | SOC-AMO/rotation | $0.09 \pm 0.03$ | N | | 0.03 | $1 \pm 4$ | N | | −0.11 | −0.08 |
| | SOC-AMO/stationary | $0.08 \pm 0.02$ | | | 0.16 | $5 \pm 6$ | | | −0.09 | 0.07 |


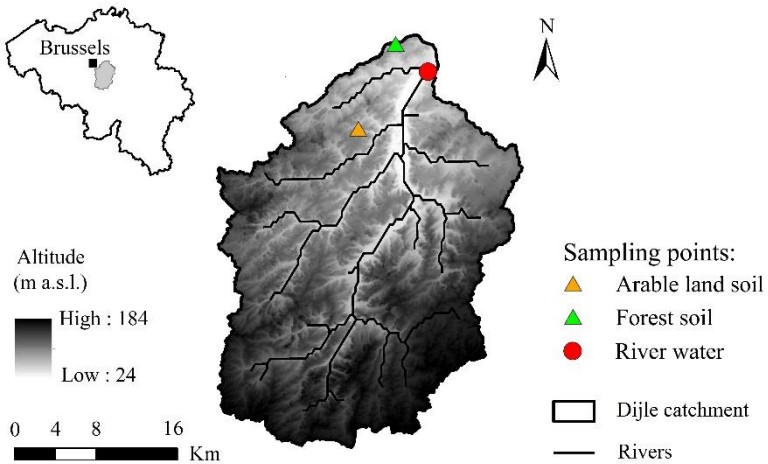

**Figure 1.** Location of Dijle catchment and the river water and soil sampling locations

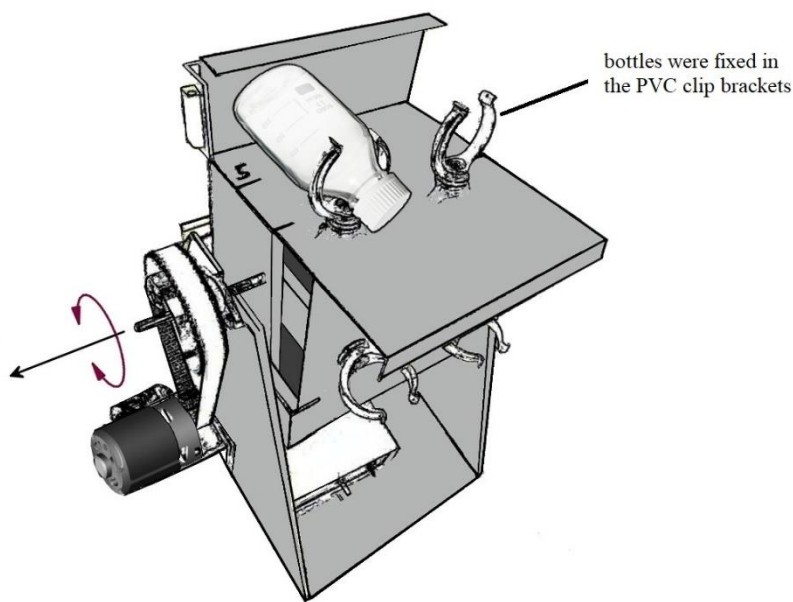

**Figure 2.** Sketch of the swing system. The swing systems were placed in a temperature-controlled (20°C) incubation room


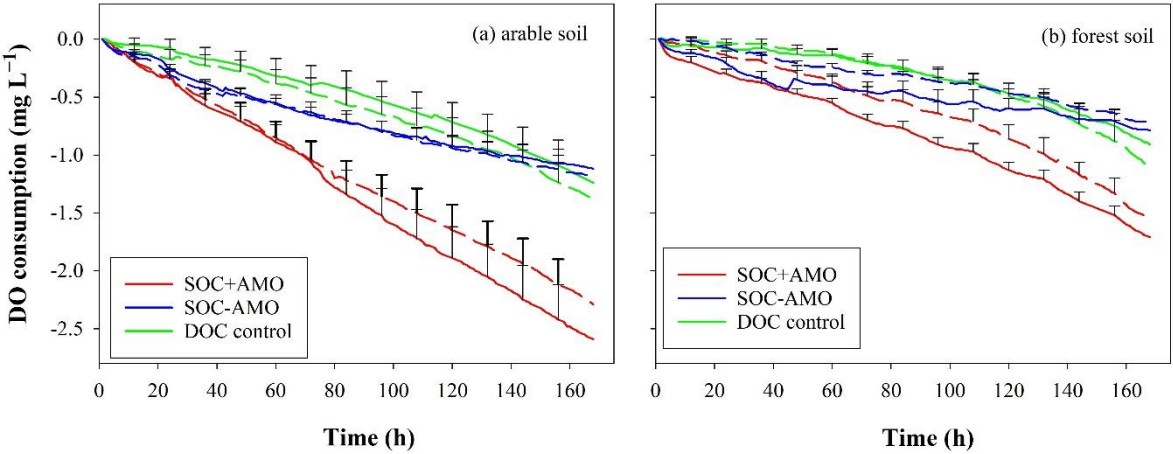

**Figure 3.** DO consumption of the SOC+AMO treatment (red), the SOC-AMO treatment (blue) and the DOC control treatment (green) during the incubations, under rotating (full lines) and stationary (dashed lines) conditions: (a) incubation with arable soil (b) incubation with forest soil. Standard errors were calculated based upon 6 repetitions with different river water samples.


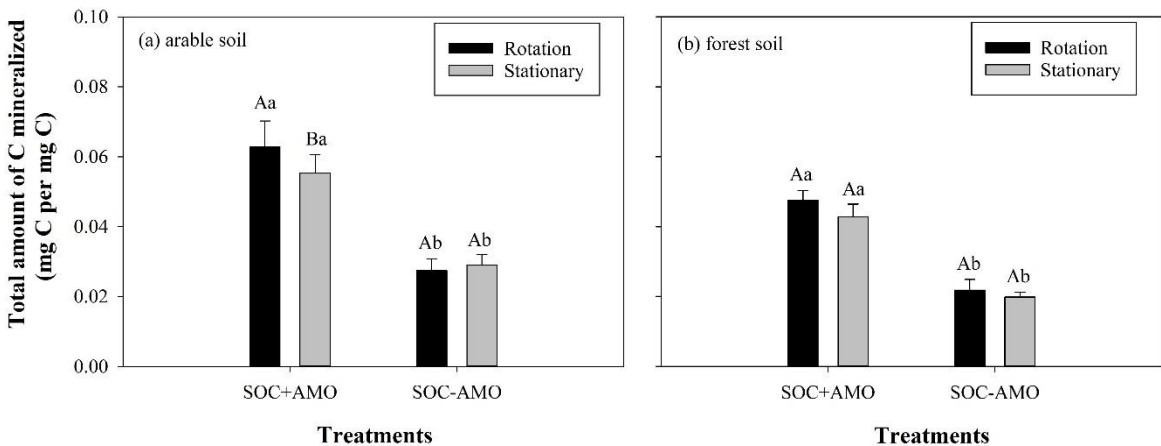

**Figure 4.** Total amount of C mineralized during the incubations with and without AMO under rotation and stationary conditions with (a) arable soil (b) forest soil. Different capital letters indicate significant differences between treatments under rotation and stationary conditions at $p < 0.05$; different lowercase letters indicate significant differences between treatments with and without AMO at $p < 0.01$.


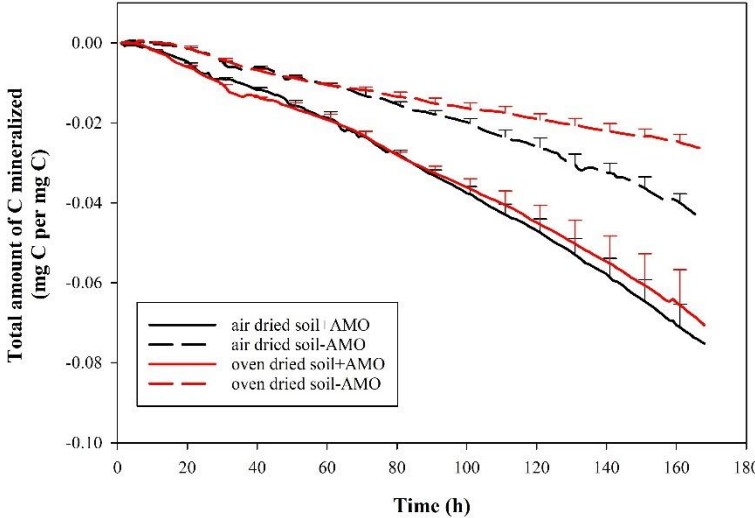

**Figure 5.** Cumulative amount of C mineralized during incubation with oven dried soil (red) and air dried soil (black), with addition of AMO (full line) and without addition of AMO (dashed line). Standard errors were calculated based upon 3 repetitions with different river water samples.


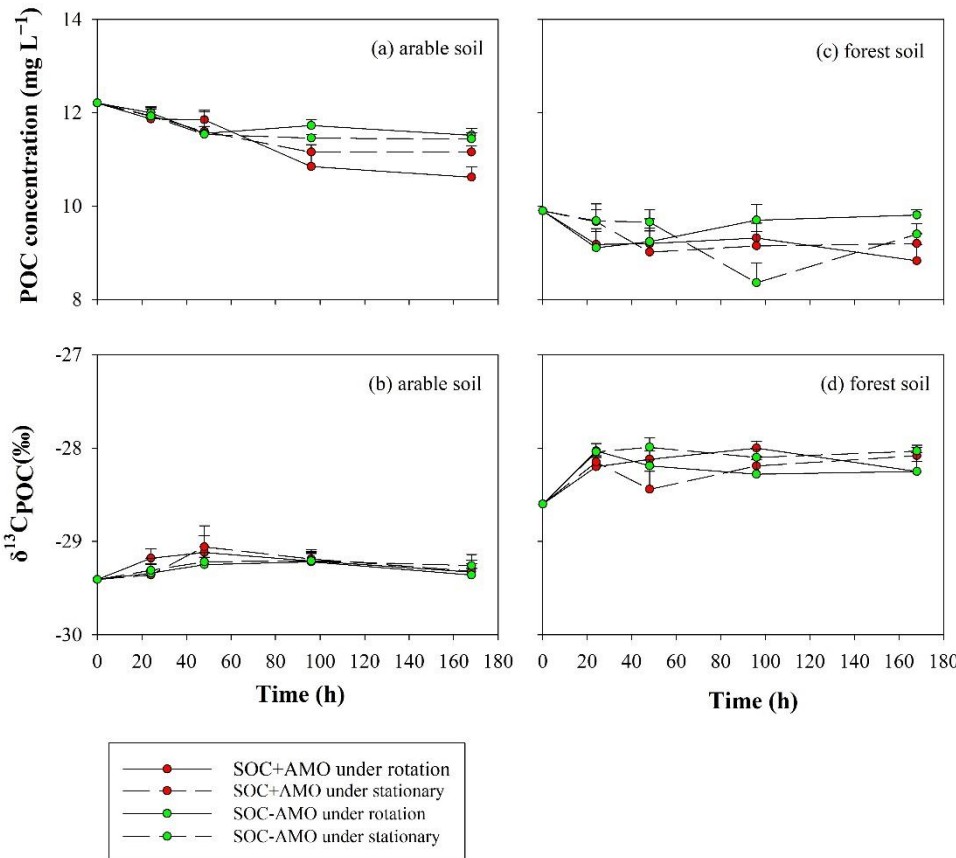

**Figure 6**. Evolution of POC concentrations and $\delta^{13}C_{POC}$ values during the incubations for the arable soil (a and b respectively) and the forest soil (c and d respectively).


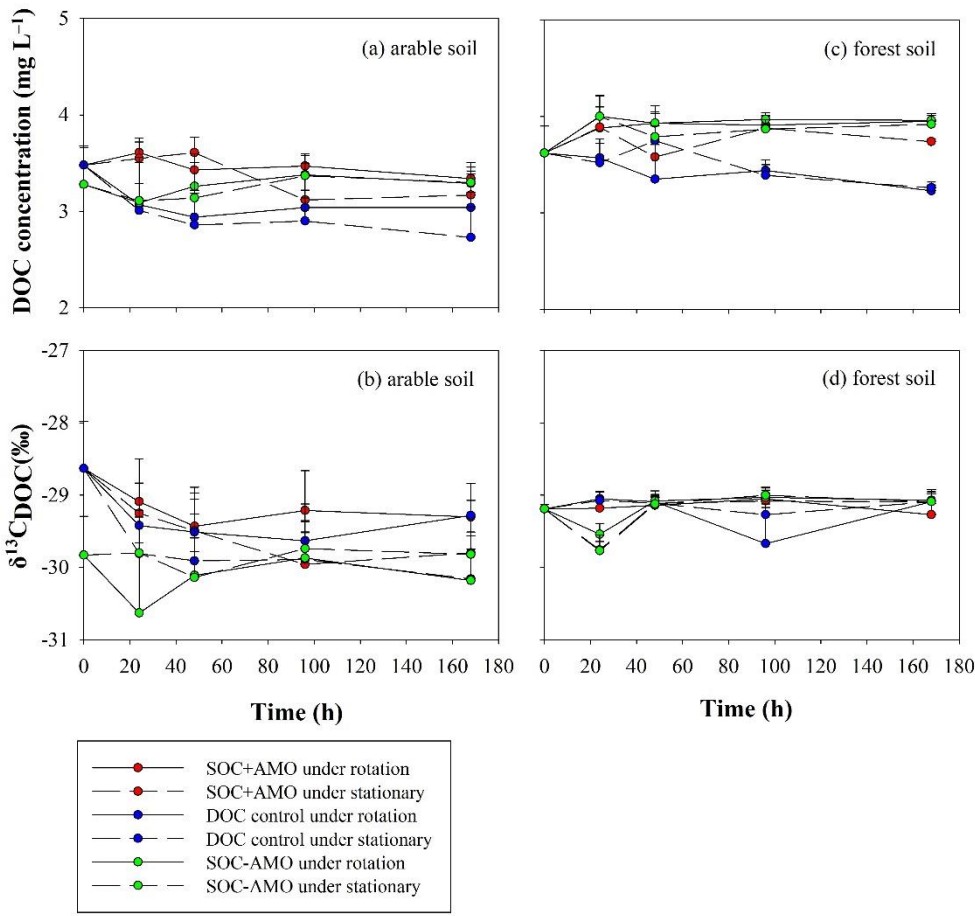

**Figure 7.** Evolution of DOC concentrations and $\delta^{13}C_{DOC}$ values during the incubations for the arable soil (a and b respectively) and the forest soil (c and d respectively).


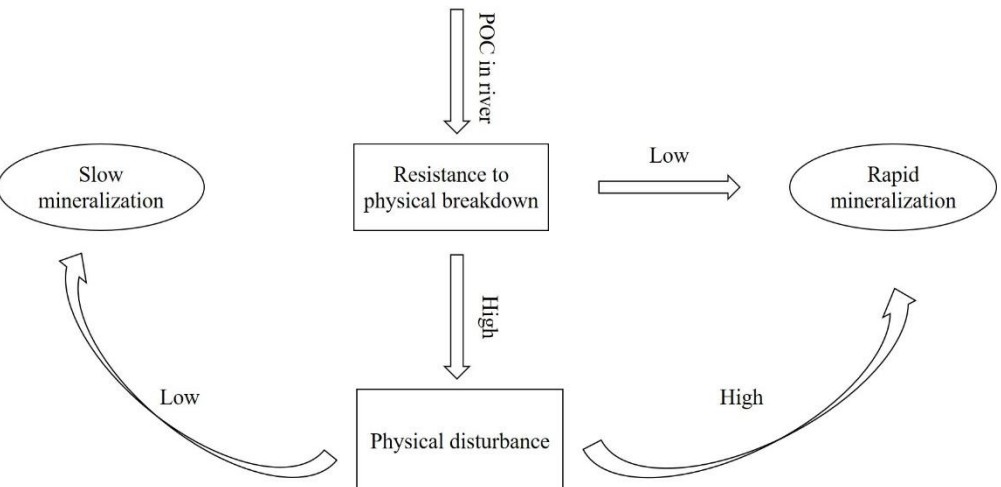

**Figure 8.** Conceptual figure representing the effect of physical disturbance on SOC decomposition upon entering river water