# Peer review of "Rapid soil organic carbon decomposition in river systems: effects of the aquatic microbial community and hydrodynamical disturbance"

_Biogeosciences, 2020_

## Referee Comment (RC1) · Anonymous Referee #1 · 17 Aug 2020

General Comments:

This study examines microbial and hydrodynamic mechanisms for soil organic carbon (SOC) in rivers based on a series of incubation experiments. The topic is both timely and important. The authors found that amending incubations with aquatic microbes drove a significant increase in SOC decomposition, whereas shaking the incubations did not have a significant impact on respiration rates.

In general the manuscript is well-written. The discussion section can be improved with a bit more referencing in parts, and also a bit deeper discussion of factors that may lead to the observed results. The introduction could also be improved by adding some nuance to the discussion about SOC. The authors overly simplify that SOC mobilized into rivers is 1) generally old and 2) assume that all soil-derived OC enters rivers in the

particulate phase. In terms of data, it would be useful to describe the starting conditions of the incubations. Specifically, the DO and nutrient levels are not mentioned, both of which could significantly influence interpretations of the results. Specific comments are given below.

Specific Comments:

Line 29: In general, the statement that much of the SOC transported into rivers is old should be made with more nuance. For example, the Mayorga reference shows that DOM in the Amazon and DIC are both predominantly modern in age. The point that tropical, temperate, and high latitude rivers behave differently is important to make.

Likewise, when reading the intro I get the impression that the authors refer to SOC as being in the river in the particulate phase only. Soil organic carbon is also leached into rivers in the dissolved phase, and most studies indicate that DOM is the main substrate fueling respiration.

Line 90: Please provide more detail. Were soil cores collected or surface soils? How deep were the soils collected from? Do the authors suspect that the SOC used was old or modern per my comment on Line 29?

Line 93: Why was this concentration chosen? That is a rather high POC concentration. Was the intention to mimic conditions you might find in the rivers being studied?

Line 105: What was the DO measurement frequency?

Line 180: Is there any nutrient data available for the experiments and/or the river water that was used? For example, more N was presumably added for the arable soils since the C:N ratio is lower than the forest soils. Nutrient limitation could be one important factor, but I am unable to evaluate this.

Line 193: Use consistent units

Line 212: Perhaps you could expand on this discussion a bit more. The Ward 2018

experiment was fundamentally different for many reasons, so it's not surprising that the results did not show the same thing. The biggest factor is that they used raw (unfiltered) water, which means the abundance and composition of POC, DOC, microbes was the same as ambient conditions, whereas this present study used an inoculum and manipulated soil additions. Another difference to mention is that the Ward experiment took place in a tropical river known for its high respiration rates as opposed to this study taking place in a temperate environment. This present study also added ∼2-3 times more SOC than is present in the turbid Amazon River and also added beads to the incubation. Do POC concentrations in the Dilje River ever get that high, i.e. were the manipulations realistic? How full of beads were the containers? This particle surface, could allow microbes to be active throughout the entire bottle even when stationary, e.g. the hypothesis by Ward 2018 was that "The relationship between rotational velocity and respiration rates exists because of the importance of interactions between suspended particles, dissolved constituents, and free-living and particle-bound microbes in driving aquatic metabolism." In contrast, this present study hypothesizes that the physical breakdown of SOC particles by disturbance is what should cause higher respiration rates.

Another important point is context about nutrient conditions. How do we know that nutrients weren't limiting?

One finding in the Ward et al 2019 paper that was cited was that respiration rates varied in response to the proportion of turbid vs clearwater river water added to incubations. There was an optimal mixture that resulted in the highest rates, and in those experiments there wasn't always a significant difference between stationary and spinning chambers. That could perhaps be something to bring up here, e.g. perhaps you would have seen different results with lower POC concentrations more similar to what you'd observe in situ. And likewise, removing the beads could have made the rotation treatment more important.

Line 298: How do you know the SOC in this experiment was old?

Lines 225-280: This section could use more references and literature comparisons.

Table 3: The caption is a little confusing. By "weight" do you mean the mass of C added?

[Figure]

---

## Referee Comment (RC2) · Anonymous Referee #2 · 17 Sep 2020

In the present study, the authors aim to understand the mechanisms of the SOC decomposition in river systems. Their study is based on 2 hypotheses (e.g., (i) in the river water, SOC is exposed to an aquatic microbial community which may be able to metabolize SOC much more quickly than the soil microbial community, and (ii) SOC decomposition in rivers is facilitated due to the hydrodynamic disturbance of sediment) for which they will investigate their weight through an incubation experiment. The paper is interesting and the study is well designed. However, before acceptance, I would suggest addressing my comments.

My main concern is related to the discussion. I think the authors could improve the discussion with a deeper exploration of their results. Some parts look weak and not necessarily well supported by the literature (see my specific comments). Then, some

other parts are the opposite. I would also suggest adding sub-sections to the Discussion to give a framework to the discussion.

Specific comments

Line 65-103: I would suggest re-organizing these three sub-sections. For example, in the first sub-section, you present the site but you also include extra information in subsection 3. Then, in reading the sub-section 2, several questions came up in my mind. But I could find the answers only on sub-section 3. In the current form, it is a bit confusing and the readers need to go back and forth to gather all the information.

Figures 6 and 7: I would suggest changing the scale of the Y-axis and/or use color. In particular, Figure 7 a and c or even d are not easily readable.

Line 172-174: Does this "increase" really mean something?

Line 205-207: This needs to be supported by the literature. Please refer to Ward et al., 2019, Wu et al., 2018, etc.

Line 244: What about the combined effect of AMO and rotation?

Line 273: After 160h of incubation, can we expect a significant shift of the $\delta$13CPOC? This needs to be discussed.

Line 278-290: You never discuss the combined effect of the occurrence of aquatic microbial organisms and physical disturbance. The discussion needs to be improved regarding this point.

Line 184-290: I have observed specific behavior for each type of soils. I think this point needs to be highlighted and also discussed. How do you explain these variations?

References:

Ward, N.D., Morrison, E.S., Liu, Y., Rivas-Ubach, A., Osborne, T.Z., Ogram, A. V., Bianchi, T.S., 2019. Marine microbial community responses related to wetland

carbon mobilization in the coastal zone. Limnol. Oceanogr. Lett. 4, 25–33. https://doi.org/10.1002/lol2.10101

Wu, X., Wu, L., Liu, Y., Zhang, P., Li, Q., Zhou, J., Hess, N.J., Hazen, T.C., Yang, W., Chakraborty, R., 2018. Microbial interactions with dissolved organic matter drive carbon dynamics and community succession. Front. Microbiol. 9, 1234. https://doi.org/10.3389/fmicb.2018.01234

---

## Author Comment (AC1) · 13 Nov 2020

**Response to Anonymous Referee #1**

We gratefully thank the referee for his/her constructive comments and have revised the manuscript accordingly. In our response below, referee comments are shown in italicized *blue*, our response in **black.**

**General Comments:**

*This study examines microbial and hydrodynamic mechanisms for soil organic carbon (SOC) in rivers based on a series of incubation experiments. The topic is both timely and important. The authors found that amending incubations with aquatic microbes drove a significant increase in SOC decomposition, whereas shaking the incubations did not have a significant impact on respiration rates.*

*In general the manuscript is well-written. The discussion section can be improved with a bit more referencing in parts, and also a bit deeper discussion of factors that may lead to the observed results. The introduction could also be improved by adding some nuance to the discussion about SOC. The authors overly simplify that SOC mobilized into rivers is 1) generally old and 2) assume that all soil-derived OC enters rivers in the particulate phase. In terms of data, it would be useful to describe the starting conditions of the incubations. Specifically, the DO and nutrient levels are not mentioned, both of which could significantly influence interpretations of the results. Specific comments are given below.*

**Reply:** We thank the reviewer for the encouraging words and constructive comments, which have been very helpful to improve the quality of the manuscript. We made a point-to-point response to the comments given above.

- We agree that the Discussion section could be improved with a deeper discussion and more thorough referencing. In doing so we will improve the Discussion section by further scrutinizing the results and connecting them with the previous studies.

- As for the introduction section, we agree that by only focusing on the particulate phase, we oversimplified the SOC mobilization in river systems. We will add more discussion on the SOC including both the particulate and the dissolved phase and the variability among rivers in different climate zones. However, we would also like to point out that the paper does indeed focus on the fate of POC in riverine environments.

- For the starting conditions of the incubations, we will add the DO concentrations for both soil types in the Result section. As for the nutrient data, we have compiled relevant data on inorganic nutrients in the section of the Dijle River where the samples were taken (Table 1, data available online from the Flanders Environment Agency), and will discuss the nutrient status in the revised version – based on these data we do not expect that inorganic nutrient concentrations would have been a limiting during our incubations.

**Table 1** The inorganic nutrients ($NH_4$, $NO_3$, $PO_4$) concentrations in the water sampling site of the Dijle river

| Year | Number of measurements | $NH_4$ ($\mu mol\ L^{-1}$) | $NO_3$ ($\mu mol\ L^{-1}$) | $PO_4$ ($\mu mol\ L^{-1}$) |
|------|------|------|------|------|
| 2020 | 3 | $24 \pm 7$ | $717 \pm 55$ | no data |
| 2016 | 6 | $48 \pm 30$ | $410 \pm 15$ | $6 \pm 1$ |
| 2014 | 6 | $26 \pm 11$ | $537 \pm 94$ | $6 \pm 1$ |

**Note**: values are Mean ± SD.

*Line 29: In general, the statement that much of the SOC transported into rivers is old should be made with more nuance. For example, the Mayorga reference shows that DOM in the Amazon and DIC are both predominantly modern in age. The point that tropical, temperate, and high latitude rivers behave differently is important to make.*

*Likewise, when reading the intro I get the impression that the authors refer to SOC as being in the river in the particulate phase only. Soil organic carbon is also leached into rivers in the dissolved phase, and most studies indicate that DOM is the main substrate fueling respiration.*

**Reply:** Agreed, this part will be improved as outlined above. Both soil samples are currently being sent out for $^{14}$C dating, these results will be added to the description of the soils used in our experiments, and we will highlight the role of climate and other environmental conditions as controls on (i) the amount and nature of POC in a river and (ii) rates of mineralisation.

*Line 90: Please provide more detail. Were soil cores collected or surface soils? How deep were the soils collected from? Do the authors suspect that the SOC used was old or modern per my comment on Line 29?*

**Reply:** Details about soils collection will be added in the Materials & Methods part - data on the $^{14}$C age will be added in the revised version (see above).

In this study, we collected surface soils: for arable land soils, we collected soil from a depth of 0-20 cm depth; for forest soil, we first removed the top leaves layer, then collected soil from a depth of 0-20 cm depth.

*Line 93: Why was this concentration chosen? That is a rather high POC concentration. Was the intention to mimic conditions you might find in the rivers being studied?*

**Reply:** Our intention was to use realistic concentrations, but we also had to make sure that effects could be detected. Before starting these incubations, we set up a series of test experiments with different POC concentrations (2 mg L$^{-1}$, 4 mg L$^{-1}$, 7 mg L$^{-1}$, 12 mg L$^{-1}$) and measured the DO consumption continuously for 7 days. The results showed that, in order to reliably detect POC mineralisation a minimum POC concentration of 12 mg L$^{-1}$ was advisable. Therefore, in order to have a detectable DO consumption from POC, POC concentration was controlled at 10–12 mg L$^{-1}$.

Furthermore, the concentrations we used mimic conditions we may find in the Dijle river during high water stages. We collected river water samples throughout a year, and POC concentrations ranged from 0.5-18.0 mg L$^{-1}$, the higher POC concentrations did not occur frequently but were observed several times during or after large rainfall events. These concentrations thus fall within the range of those observed in our study area and are not uncommon in various other (turbid) river systems.

*Line 105: What was the DO measurement frequency?*

**Reply:** DO was measured using an optical oxygen meter (FireStingO$_2$) with a measurement frequency of 10 s. This information will be included in the revised manuscript.

*Line 180: Is there any nutrient data available for the experiments and/or the river water that was used? For example, more N was presumably added for the arable soils since the C:N ratio is lower than the forest soils. Nutrient limitation could be one important factor, but I am unable to evaluate this.*

**Reply:** See reply to the previous suggestions: data on inorganic nutrients (NH$_4^+$, NO$_3^-$, and phosphate) in the river water are collected by the Flanders Environment Agency, and available from a publicly accessible dataportal. We will summarize these data in the revised manuscript (Table 1).

*Line 193: Use consistent units*

**Reply:** Consistent units will be used in the revised manuscript.

*Line 212: Perhaps you could expand on this discussion a bit more. The Ward 2018 experiment was fundamentally different for many reasons, so it's not surprising that the results did not show the same thing. The biggest factor is that they used raw (unfiltered) water, which means the abundance and composition of POC, DOC, microbes was the same as ambient conditions, whereas this present study used an inoculum and manipulated soil additions. Another difference to mention is that the Ward experiment took*

*place in a tropical river known for its high respiration rates as opposed to this study taking place in a temperate environment. This present study also added ~2-3 times more SOC than is present in the turbid Amazon River and also added beads to the incubation. Do POC concentrations in the Dilje River ever get that high, i.e. were the manipulations realistic? How full of beads were the containers? This particle surface, could allow microbes to be active throughout the entire bottle even when stationary, e.g. the hypothesis by Ward 2018 was that "The relationship between rotational velocity and respiration rates exists because of the importance of interactions between suspended particles, dissolved constituents, and free-living and particle-bound microbes in driving aquatic metabolism." In contrast, this present study hypothesizes that the physical breakdown of SOC particles by disturbance is what should cause higher respiration rates.*

**Reply:** we agree that the differences in experimental conditions might indeed influence POC decomposition rates, we will bring this aspect into the discussion in the revised version. However, these differences are *as such* not a reason to expect that shaking would not play a role here: the factors described above as the reasons why shaking can be important are also present in our experiments. Thus, there must be an additional factor. Our hypothesis is that soil/aggregate strength is also crucial.

*Another important point is context about nutrient conditions. How do we know that nutrients weren't limiting?*

*One finding in the Ward et al 2019 paper that was cited was that respiration rates varied in response to the proportion of turbid vs clearwater river water added to incubations. There was an optimal mixture that resulted in the highest rates, and in those experiments, there wasn't always a significant difference between stationary and spinning chambers. That could perhaps be something to bring up here, e.g. perhaps you would have seen different results with lower POC concentrations more similar to what you'd observe in situ. And likewise, removing the beads could have made the rotation treatment more important.*

**Reply:** We agree that POC concentrations and the beads additions may play a role when exam the difference between stationary and rotation condition, we will discuss the potential effect of these factors on POC decomposition rates in the revised manuscript. However, as outlined above, we still believe these could not fully explain the minor effect of rotation in our study, which lead us to hypothesized that soil/aggregate strength might play an important role.

As for the nutrient conditions, we have compiled the inorganic nutrient in the section of the Dijle River where the samples were taken as outlined above. We will discuss the nutrient status in the revised manuscript.

*Line 298: How do you know the SOC in this experiment was old?*

**Reply:** See earlier replies: the 14C age of the bulk organic C in our two soils is currently being analysed, we will report these results in the revised version.

*Lines 225-280: This section could use more references and literature comparisons.*

**Reply:** As outlined above, we will seriously rework on the Discussion section by further exploring the results and connecting them with the previous studies.

*Table 3: The caption is a little confusing. By "weight" do you mean the mass of C added?*

**Reply:** Yes, by "weight" we mean the mass of C added in the beginning and recovered from the final sample. "weight" will be replaced by "the mass of C" in the revised manuscript.

---

## Author Comment (AC2) · 13 Nov 2020

**Response to Anonymous Referee #2**

We thank Referee #2 for his/her constructive comments, which are addressed as explained below. In our response below, referee comments are shown in italicized *blue*, our response in **black**.

**General comments**

*In the present study, the authors aim to understand the mechanisms of the SOC decomposition in river systems. Their study is based on 2 hypotheses (e.g., (i) in the river water, SOC is exposed to an aquatic microbial community which may be able to metabolize SOC much more quickly than the soil microbial community, and (ii) SOC decomposition in rivers is facilitated due to the hydrodynamic disturbance of sediment) for which they will investigate their weight through an incubation experiment. The paper is interesting and the study is well designed. However, before acceptance, I would suggest addressing my comments.*
*My main concern is related to the discussion. I think the authors could improve the discussion with a deeper exploration of their results. Some parts look weak and not necessarily well supported by the literature (see my specific comments). Then, some other parts are the opposite. I would also suggest adding sub-sections to the Discussion to give a framework to the discussion.*

**Reply:** We thank the reviewer for the constructive comments on the Discussion section, which will be addressed accordingly in the manuscript. We will seriously rework on the Discussion section by (i) further exploring the results and connecting them with previous studies, (ii) improve the structure and provide a better framework.

**Specific comments**

*Line 65-103: I would suggest re-organizing these three sub-sections. For example, in the first sub-section, you present the site but you also include extra information in subsection 3. Then, in reading the sub-section 2, several questions came up in my mind. But I could find the answers only on sub-section 3. In the current form, it is a bit confusing and the readers need to go back and forth to gather all the information.*

**Reply:** Thank you for this constructive comment, we will reorganize the subsections to better describe the approach in the revised version.

*Figures 6 and 7: I would suggest changing the scale of the Y-axis and/or use colour. In particular, Figure 7 a and c or even d are not easily readable.*

**Reply:** Agreed, we will improve the figures in the revised manuscript.

*Line 172-174: Does this "increase" really mean something?*

**Reply:** Yes, this increase was consistent in all replicates, we think that the increase of $\delta^{13}C_{POC}$ values during the first 24–48 hours suggests that during this period an isotopically lighter POC fraction was preferentially mineralised. This resulted in the POC in the aquatic environment becoming enriched in $^{13}C$ compared to the POC in the original soil sample. While this shift in $\delta^{13}C$ values is relatively small, we do feel it is significant given that it is consistent in both experiments, and larger than the analytical error, However, we are careful in our discussion and as to avoid any overinterpretations on this.

*Line 205-207: This needs to be supported by the literature. Please refer to Ward et al., 2019, Wu et al., 2018, etc.*

**Reply:** Agreed, this part will be improved following the suggestions made.

*Line 244: What about the combined effect of AMO and rotation?*

**Reply:** To identify the combined effect of AMO and rotation on the C decomposition rates, two-way ANOVA with the presence of AMO and disturbance conditions as the main factors was employed for the two soil types, separately. Results showed that the presence of AMO and rotation had no significant combined effect on the C decomposition rates for both soil types in our study

(*arable land: p=0.430; forest: p=0.683*). This is not surprising, as we proposed in the conceptual model, the mere immersion of soil particles in water might be sufficient to destroy most of the soil particles which were loess derived with the low structure stability. Therefore, further disturbance did not significantly increase the interactions between soil particles and microbial organisms. This combined effect might be more evident for SOC with strong physical protection.

*Line 273: After 160h of incubation, can we expect a significant shift of the $\delta^{13}C_{POC}$? This needs to be discussed.*

**Reply:** If this mineralization does not selectively affect specific fractions of the POC pool, the $\delta^{13}C_{POC}$ values can be expected to remain more or less constant throughout the incubation period. We will add discussion regarding to this aspect in the revised manuscript.

*Line 278-290: You never discuss the combined effect of the occurrence of aquatic microbial organisms and physical disturbance. The discussion needs to be improved regarding this point.*

**Reply:** The combined effect of the occurrence of aquatic microbial organisms and physical disturbance has been outlined above. We agreed that the combined effect of the occurrence of AMO and physical disturbance would be an interesting point to be further studied, and we will bring this into discussion in the revised manuscript.

*Line 184-290: I have observed specific behaviour for each type of soils. I think this point needs to be highlighted and also discussed. How do you explain these variations?*

**Reply:** We thank the referee for their constructive comment on comparison of the two soil types. We feel that the different SOC contents and the nature of the SOC (derived from agricultural crops versus forest litter) could offer a likely explanation for, the observed difference in decomposition behaviour. We will add a paragraph in the Discussion section to compare the decomposition behaviour, and possibly link this to the $^{14}C$ ages which are currently being determined.

---

## Author Response (AR1)

**Response to Anonymous Referee #1**

We gratefully thank the referee for his/her constructive comments and have revised the manuscript accordingly. In our response below, referee comments are shown in italicized *blue*, our response in **black.** Please note that the line and page numbers in our responses refer to the revised version of the manuscript.

5 **General Comments:**

*This study examines microbial and hydrodynamic mechanisms for soil organic carbon (SOC) in rivers based on a series of incubation experiments. The topic is both timely and important. The authors found that amending incubations with aquatic microbes drove a significant increase in SOC decomposition, whereas shaking the incubations did not have a significant impact on respiration rates.*

10 *In general the manuscript is well-written. The discussion section can be improved with a bit more referencing in parts, and also a bit deeper discussion of factors that may lead to the observed results. The introduction could also be improved by adding some nuance to the discussion about SOC. The authors overly simplify that SOC mobilized into rivers is 1) generally old and 2) assume that all soil-derived OC enters rivers in the particulate phase. In terms of data, it would be useful to describe the starting conditions of the incubations. Specifically, the DO and nutrient levels are not mentioned, both of which could significantly influence*

15 *interpretations of the results. Specific comments are given below.*

**Reply:** We thank the reviewer for the encouraging words and constructive comments, which have been very helpful to improve the quality of the manuscript. We have revised the manuscript according to the comments given above.

    i) We agree that the Discussion section could be improved with a deeper discussion and more thorough referencing. In doing so we have improved the Discussion section by further scrutinizing the results and connecting them with the previous

20         studies. For more detailed information, see **Line 225-227, Line 237-245, Page 7; Line 305-309, Line 312-315, Page 9**.

    ii) As for the introduction section, we agree that by only focusing on the particulate phase, we oversimplified the SOC mobilization in river systems. We have added discussion on the SOC including both the particulate and the dissolved phase and the variability among rivers (see **Line 27-31, Page 1**). However, our experiments were specifically designed to examine the fate of POC in riverine environment.

25     iii) For the starting conditions of the incubations, we have added the initial DO concentration of the river water (varied between 7.58-10.67 mg L$^{-1}$) in the Results section (see **Line 146, Page 5**). As for the nutrient data, we have compiled relevant data on inorganic nutrients in the section of the Dijle River where the samples were taken (Table 2, data available online from the Flanders Environment Agency)–based on these data we do not expect that DO or inorganic nutrient concentrations would have been a limiting during our incubations (**Table 2**). This has been discussed in the Discussion

30         section, see **Line 312-315, Page 9.**

**Table 2.** Nutrient concentrations in the Dijle river water

| Year | Number of measurements | NH$_4^+$ (µmol L$^{-1}$) | NO$_3^-$ (µmol L$^{-1}$) | PO$_4^{3-}$ (µmol L$^{-1}$) |
|------|------------------------|--------------------------|--------------------------|------------------------------|
| 2020 | 3 | $24 \pm 7$ | $717 \pm 55$ | no data |
| 2016 | 6 | $48 \pm 30$ | $410 \pm 15$ | $6 \pm 1$ |
| 2014 | 6 | $26 \pm 11$ | $537 \pm 94$ | $6 \pm 1$ |

**Note**: values are Mean ± SD.

*Line 29: In general, the statement that much of the SOC transported into rivers is old should be made with more nuance. For example, the Mayorga reference shows that DOM in the Amazon and DIC are both predominantly modern in age. The point that tropical, temperate, and high latitude rivers behave differently is important to make.*

*Likewise, when reading the intro I get the impression that the authors refer to SOC as being in the river in the particulate phase only. Soil organic carbon is also leached into rivers in the dissolved phase, and most studies indicate that DOM is the main substrate fueling respiration.*

**Reply:** Agreed, this part has been improved as outlined above.

  i)   We have added more discussion on the SOC including both the particulate and the dissolved phase and the variability among rivers, see **Line 27-31, Page 1**.

  ii)  Data on the [14]C age has been added to the description of the soils used in our experiments (**Table 3**).

  iii) In addition, we reanalysed soil texture, and the results have been corrected in the revised manuscript (**Table 3**).

**Table 3.** Characteristics of the two soils used in this study. Texture was determined using laser diffraction (Coulter LS 13 320).

| Soil samples | soil texture | | OC content (%) | C/N (weight/weight) | $\delta^{13}$C (‰) | [14]C age (yr BP) |
|---|---|---|---|---|---|---|
| arable land | Sand % | 23 | 2.40 | 9.9 | −29.4 | $267 \pm 21$ |
| | Loam % | 69 | | | | |
| | Clay % | 8 | | | | |
| forest | Sand % | 39 | 5.20 | 17.6 | −28.6 | $334 \pm 22$ |
| | Loam % | 56 | | | | |
| | Clay % | 5 | | | | |

*Line 90: Please provide more detail. Were soil cores collected or surface soils? How deep were the soils collected from? Do the authors suspect that the SOC used was old or modern per my comment on Line 29?*

**Reply:** Details about soil collection have been added in the Materials & Methods part, and data on the [14]C age has been added in the revised version (**Table 3**). In this study, we collected surface soils: for arable land soils, we collected soil from a depth of 0-20 cm depth; for forest soil, we first removed the top litter layer, then collected soil from a depth of 0-20 cm depth. For more detailed information, see **Line 93-94, Page 3**.

*Line 93: Why was this concentration chosen? That is a rather high POC concentration. Was the intention to mimic conditions you might find in the rivers being studied?*

**Reply:** Our intention was to use realistic concentrations, but we also had to ensure that effects could be detected. Before starting these incubations, we set up a series of trial experiments with different POC concentrations (2 mg L$^{-1}$, 4 mg L$^{-1}$, 7 mg L$^{-1}$, 12 mg L$^{-1}$) and measured the DO consumption continuously for 7 days. The results showed that, in order to reliably detect POC mineralisation a minimum POC concentration of 12 mg L$^{-1}$ was more sensitive. We therefore decide to conduct further experiments with a POC concentration of around 10–12 mg L$^{-1}$.

Furthermore, the concentrations we used are well within the range of conditions we may find in the Dijle river during high water stages. We collected river water samples throughout a year, and POC concentrations ranged from 0.5-18.0 mg L$^{-1}$, the higher POC concentrations did not occur frequently but were observed several times during or after large rainfall events (when the inputs of soil-derived organic carbon can be assumed to dominate). These concentrations thus fall within the range of those observed in our study area and are not uncommon in various other (turbid) river systems.

65     **Reply:** DO was measured using an optical oxygen meter (FireStingO$_2$) with a measurement frequency of 10 s. This information has been included in the revised manuscript, see **Line 110, Page 4**.

70     **Reply:** See reply to the previous suggestions: data on inorganic nutrients (NH$_4^+$, NO$_3^-$, and phosphate) in the river water are collected by the Flanders Environment Agency, and available from a publicly accessible dataportal. We have summarized these data and added it as **Table 2** in the revised manuscript. Given these concentrations, we assume that nutrient limitations are unlikely. This has been added to the Discussion section, for more detailed information, see **Line 312-315, Page 9.**

75     **Reply:** This has been checked and corrected as suggested, see **Line 197, Page 6**.

80

85

*Line 212: Perhaps you could expand on this discussion a bit more. The Ward 2018 experiment was fundamentally different for many reasons, so it's not surprising that the results did not show the same thing. The biggest factor is that they used raw (unfiltered) water, which means the abundance and composition of POC, DOC, microbes was the same as ambient conditions, whereas this present study used an inoculum and manipulated soil additions. Another difference to mention is that the Ward experiment took*

*place in a tropical river known for its high respiration rates as opposed to this study taking place in a temperate environment. This present study also added ~2-3 times more SOC than is present in the turbid Amazon River and also added beads to the incubation. Do POC concentrations in the Dilje River ever get that high, i.e. were the manipulations realistic? How full of beads were the containers? This particle surface, could allow microbes to be active throughout the entire bottle even when stationary, e.g. the hypothesis by Ward 2018 was that "The relationship between rotational velocity and respiration rates exists because of the*

*importance of interactions between suspended particles, dissolved constituents, and free-living and particle-bound microbes in driving aquatic metabolism." In contrast, this present study hypothesizes that the physical breakdown of SOC particles by disturbance is what should cause higher respiration rates.*

    **Reply:** We agree that the differences in experimental conditions might indeed influence POC decomposition rates, and we have brought this aspect into the discussion in the revised version. However, these differences are *as such* not a reason to expect that

90     shaking would not play a role here: the factors described above as a rational for shaking to be important are also present in our experiments. Thus, there must be an additional factor. Our hypothesis is that soil/aggregate strength is also crucial. For more detailed information, see **Line 225-227, Page 7.**

95

*Another important point is context about nutrient conditions. How do we know that nutrients weren't limiting?*

*One finding in the Ward et al 2019 paper that was cited was that respiration rates varied in response to the proportion of turbid*

*vs clearwater river water added to incubations. There was an optimal mixture that resulted in the highest rates, and in those experiments, there wasn't always a significant difference between stationary and spinning chambers. That could perhaps be something to bring up here, e.g. perhaps you would have seen different results with lower POC concentrations more similar to what you'd observe in situ. And likewise, removing the beads could have made the rotation treatment more important.*

    **Reply:** As outlined above, we have discussed the potential effect of the different experimental settings on POC decomposition

100     rates in the revised manuscript. However, we still believe these could not fully explain the minor effect of rotation in our study, which leads us to hypothesized that soil/aggregate strength might play an important role.

As for the nutrient conditions, we have compiled the inorganic nutrient in the section of the Dijle River where the samples were taken as outlined above (**Table 2**). The nutrient status has been discussed in the revised manuscript, see **Line 312-315, Page 9.**

*Line 298: How do you know the SOC in this experiment was old?*

105 **Reply:** As presented in **Table 3**, the radiocarbon age of the surface soil samples used in our incubation were 267-334 yr BP, which were relatively young compared to that found in other aquatic ecosystems (in the range of 1000–5000 y B.P.) (Marwick et al., 2015; Raymond and Bauer, 2001; McCallister and Del Giorgio, 2012). Given the potential variability of the radiocarbon age of SOC among rivers, we have improved our manuscript without highlighting old SOC in the revised manuscript, see **Line 7, Line 32, Page 1; Line 255, Page 7; Line 327, Page 9**.

110 **Reference**

Marwick, T. R., Tamooh, F., Teodoru, C. R., Borges, A. V, Darchambeau, F. and Bouillon, S.: The age of river-transported carbon: a global perspective, Global Biogeochem. Cycles, 29, 122–137, doi:10.1002/2014GB004911.Received, 2015.

McCallister, S. L. and Del Giorgio, P. A.: Evidence for the respiration of ancient terrestrial organic C in northern temperate lakes and streams, Proc. Natl. Acad. Sci. U. S. A., 109(42), 16963–16968, doi:10.1073/pnas.1207305109, 2012.

115 Raymond, P. A. and Bauer, J. E.: Riverine export of aged terrestrial organic matter to the North Atlantic Ocean, Nature, 409(6819), 497–500, doi:10.1038/35054034, 2001.

*Lines 225-280: This section could use more references and literature comparisons.*

Reply: As outlined in the former reply to the general comments, we have reworked the Discussion section by further exploring the
120 results and connecting them with previous studies. For more detailed information, see **Line 225-227, Line 237-245, Page 7; Line 305-309, Line 312-315, Page 9.**

*Table 3: The caption is a little confusing. By "weight" do you mean the mass of C added?*

**Reply:** Yes, by "weight" we mean the mass of C added in the beginning and recovered from the final sample. "weight" has been replaced by "the mass of C" in the caption of Table 4, see **Line 472-473, Page 17**.

125

**Response to Anonymous Referee #2**

We thank Referee #2 for his/her constructive comments, which are addressed as explained below. In our response below, referee comments are shown in italicized *blue*, our response in **black**.

**General comments**

*In the present study, the authors aim to understand the mechanisms of the SOC decomposition in river systems. Their study is based on 2 hypotheses (e.g., (i) in the river water, SOC is exposed to an aquatic microbial community which may be able to metabolize SOC much more quickly than the soil microbial community, and (ii) SOC decomposition in rivers is facilitated due to the hydrodynamic disturbance of sediment) for which they will investigate their weight through an incubation experiment. The paper is interesting and the study is well designed. However, before acceptance, I would suggest addressing my comments.*

*My main concern is related to the discussion. I think the authors could improve the discussion with a deeper exploration of their results. Some parts look weak and not necessarily well supported by the literature (see my specific comments). Then, some other parts are the opposite. I would also suggest adding sub-sections to the Discussion to give a framework to the discussion.*

**Reply:** We thank the reviewer for the constructive comments on the Discussion section, which have been addressed accordingly in the revised manuscript. We have seriously reworked the discussion by further exploring the results and connecting them with previous studies which hopefully further facilitate the reader's understanding. For more detailed information, see **Line 225-227, Line 237-245, Page 7; Line 305-309, Line 312-315, Page 9.**

**Specific comments**

*Line 65-103: I would suggest re-organizing these three sub-sections. For example, in the first sub-section, you present the site but you also include extra information in subsection 3. Then, in reading the sub-section 2, several questions came up in my mind. But I could find the answers only on sub-section 3. In the current form, it is a bit confusing and the readers need to go back and forth to gather all the information.*

**Reply:** Thank you for this constructive comment, we have reorganized the subsections to better describe the approach in the revised version. For more detailed information, see **Line 70, Page 2; Line 89, Line 100-102, Page 3**.

*Figures 6 and 7: I would suggest changing the scale of the Y-axis and/or use colour. In particular, Figure 7 a and c or even d are not easily readable.*

**Reply:** Agreed, we have changed the scale of the Y-axis and have also used the colour to better present our results in the revised manuscript (see **Figure 6, Figure 7**).

*Line 172-174: Does this "increase" really mean something?*

**Reply:** Yes, this increase was consistent in all replicates, we think that the increase of $\delta^{13}C_{POC}$ values during the first 24–48 hours suggests that during this period an isotopically lighter POC fraction was preferentially mineralised. This resulted in the POC in the aquatic environment becoming enriched in $^{13}C$ compared to the POC in the original soil sample. While this shift in $\delta^{13}C$ values is relatively small, we do feel it is significant given that it is consistent in both experiments, and larger than the analytical error, However, we are careful in our discussion and as to avoid any overinterpretations on this.

*Line 205-207: This needs to be supported by the literature. Please refer to Ward et al., 2019, Wu et al., 2018, etc.*

**Reply:** Agreed, this part have been improved following the suggestions made, see **Line 220, Page 7**.

*Line 244: What about the combined effect of AMO and rotation?*

**Reply:** To identify the combined effect of AMO and rotation on the C decomposition rates, two-way ANOVA with the presence of AMO and disturbance conditions as the main factors was employed for the two soil types, separately. Results showed that the presence of AMO and rotation had no significant combined effect on the C decomposition rates for both soil types in our study (*arable land: p=0.430; forest: p=0.683*). This is not surprising, as we proposed in the conceptual model, the mere immersion of soil particles in water might be sufficient to destroy most of the soil particles which were loess derived with the low structure stability. Therefore, further disturbance did not significantly increase the interactions between soil particles and microbial organisms. This combined effect might be more evident for SOC with strong physical protection. This insignificant combined effect has been added in the Result and Discussion section. For more detailed information, see **Line 159-160, Page 5; Line 305-309, Page 9.**

*Line 273: After 160h of incubation, can we expect a significant shift of the δ13CPOC? This needs to be discussed.*

**Reply:** If this mineralization does not selectively affect specific fractions of the POC pool, the $\delta^{13}C_{POC}$ values can be expected to remain more or less constant throughout the incubation period. This has already been discussed in the manuscript, see **Line 281-285, Page 8**.

*Line 278-290: You never discuss the combined effect of the occurrence of aquatic microbial organisms and physical disturbance. The discussion needs to be improved regarding this point.*

**Reply:** The combined effect of the occurrence of aquatic microbial organisms and physical disturbance has been outlined above. We agreed that the combined effect of the occurrence of AMO and physical disturbance would be an interesting point to be further studied, and we have brought this into discussion in the revised manuscript. For more detailed information, see **Line 305-309, Page 9.**

In addition, the statistical test to identify the combined effect of AMO and physical disturbance on the C decomposition rates has been added in the sub-section "2.3 Statistical analysis", see **Line 140-142, Page 4**.

*Line 184-290: I have observed specific behaviour for each type of soils. I think this point needs to be highlighted and also discussed. How do you explain these variations?*

**Reply:** We thank the referee for their constructive comment on comparison of the two soil types. We argue that the different SOC content and the nature of the SOC (derived from agricultural crops versus forest litter) could offer a likely explanation for the observed difference in decomposition behaviour. This has been discussed in the Discussion section to compare the decomposition behaviour, and we have also linked this to the $^{14}C$ ages data. For more detailed information, see **Line 237-245, Page 7.**

---

## Author Response (AR2)

Dear editor,

We gratefully thank you for your comments and have revised the manuscript accordingly. In our response below, your comments are shown in italicized *blue*, our response in **black.**

*1. Title: First, "decomposition", not "decomposition rate", is rapid, so it would be more appropriate if the title is corrected to "Rapid soil organic carbon decomposition in river systems…" Second, "priming" might have played a role in the enhanced decomposition of SOC, but you have not introduced the concept at all, either in the abstract or the introduction. You can use a more relevant term you used, such as "aquatic microbial communities".*

**Reply:** The title has been improved according to the comments above.

**The title now reads** "Rapid soil organic carbon decomposition in river systems: effect of the aquatic microbial community and hydrodynamical disturbance".

*2. Line 8 (track-change version) "which is often old": As the first reviewer commented (and you responded to the comment), you need to make clear that soil organic carbon is a mixture or continuum of materials varying in age. This specific sentence is confusing because it is not clear whether "which" refers to "a significant fraction" or "SOC". As your 14C data showed, the significant fraction that rapidly decomposed might be a rather young, labile fraction. Therefore, you need to clarify the sentence.*

**Reply:** This statement has been revised in the manuscript.

**Line 6-8 now reads** "Mounting evidence indicates that a significant fraction of this SOC, *which displays a wide range of ages*, is rapidly decomposed after entering the river system".

*3. Line 29 "storage time": Do you mean "retention time"? I would place "among rivers" before "depending…..".*

**Reply:** This sentence has been improved according to the comments above. See **Line 27-28, Page 1**.

**Line 27-28 now reads** "The mobilized SOC can display a very wide range of ages among rivers depending on the carbon sources and retention time".

*4. Line 99 "4 ml unfiltered river water was added": Please provide the ratio of inoculum to the incubated water sample.*

**Reply:** Agreed, this information has been added in the manuscript, see **Line 93-94, Page 3**.

**Line 93-94 now reads** "4 ml unfiltered river water (ratio of inoculum to the incubated water sample: 1:79) was added to serve as an inoculum for treatments with aquatic microbial organisms".

*5. Line 104 "POC concentration was controlled at 10–12 mg L−1": here or somewhere later in the discussion, please provide the rationale for using this rather high (but often observed during intense rainfall events) range of POC concentrations, as you described in your response to the first reviewer's comment. In addition, please provide the water sample volume to which 60 mg or 160 mg soil was added.*

**Reply:** Thank you for your constructive comments. We now have added the rationale for using this high POC concentration. For more detailed information, see **line 98-101, Page 3.**

**Line 98-101 now reads** "In order to obtain a detectable rate of oxygen consumption, the POC concentration was controlled at 10–12 mg $L^{-1}$ by adding 160 mg arable soil and 60 mg forest soil in 320 ml river water (details in Table 3). While the sediment and POC concentrations we used in our experiments are relatively high, they are not unrealistic: during high flood we observed POC concentrations exceeding 10 mg $L^{-1}$ in ca. 5% of our samples".

*6. Line 203: As mentioned earlier, only "some fraction of SOC", not all SOC components, can be mineralized quickly.*

**Reply:** Agreed, this has been revised in the manuscript, see **Line 203-204, Page 6**.

**Line 203-204 now reads** "Our results show that a fraction of the terrestrial SOC can indeed be mineralized relatively quickly when introduced in an aquatic environment".

*7. Line 205 "83–139 μg C L−1 d−1μg C−1 d−1.": This rate depends on how much POC you put in your water samples, so it would be more appropriate if you provide the values in the unit of ug C/g soil C/d.*

**Reply:** Agreed, we have normalized decomposition rates relative to the amount of POC, and have added it in the revised manuscript. We would also keep the absolute rates in this section, since we compare them here to date reported in the papers cited there. See **Line 205, Page 6**.

*8. Lines 182-187: Please indicate significant differences in the corresponding tables and figures. For instance, in Figs. 6 and 7 you can easily show this by putting asterisks on top of the corresponding values.*

**Reply:** The significances discussed in the text has been indicated in **Table 4**.

As for Figure 6 and Figure 7, since we compared (i) different soil types, (ii) rotation *vs.* stationary and (iii) with or without the presence of AMO, we think that including significant differences in Figure 6 and Figure 7 may lead to confusions. Instead, we now have mentioned all significances explicitly in the text, and the ones central to our discussion are summarized in Table 4. For more detailed information, see **Line 171-173, Line 175-177, Page 5; Line 197, Page 6**.

*9. Line 238 "priming": Please define this concept at its first use and provide a relevant reference.*

**Reply:** In this manuscript, with "priming" we mean the stimulation effect of the presence of the aquatic microbial community on SOC decomposition rates. In order to avoid leading misunderstanding on "priming effect", we have replaced "priming" with specific descriptions. See **Line 1-2, Page 1**; **Line 236-237, Page 7**.

**Line 1-2 now reads** "Rapid soil organic carbon decomposition in river systems: effect of the aquatic microbial community and hydrodynamical disturbance".

**Line 236-237 now reads** "Thus, our data do indicate that the relative role of physical disturbance vs. that of exposure to an aquatic microbial community may vary considerably between different ecosystems".

*10. Line 252 "systematic": Difficult to understand; please replace the word or elaborate what you meant.*

**Reply:** We have rephrased this sentence in the manuscript. See **Line 250-251, Page 7**.

**Line 250-251 now reads** "However, the effect of mechanical disturbance is small and statistically insignificant, also for the forest soil.".

*11. Line 255 "which is consistent with its higher 14C age (Table 3)": This is not always the case, because aged SOC tends to have lower C/N ratios given N accumulation during decomposition.*

**Reply:** Given that the age difference between the two soil types is very small (267 yr *vs.* 334 yr), we have removed this statement in the revised manuscript.

*12. Line 316 "have a weak structure": Is this a confirmed fact? You can be more cautious in wording, like using "may".*

**Reply:** This statement was made given that the soils used in this study were loess derived from a Belgian loess belt, known for their very low structure stability (Govers, 1991). We now have improved this part by adding the reference. See **Line 315-316, Page 9.**

**Reference**

Govers, G.: Rill Erosion on Arable Land in Central Belgium: Rates, Controls and Predictability, Catena, 18(2), 133-155, 10.1016/0341-8162(91)90013-n, 1991.

*13. Tables 3 & 4: "A"rable land, "F"orest*

**Reply:** These have been corrected in **Table 3 & 4**.

*14. Fig. 8: For "high", please use consistently the uppercase initial letter (H).*

**Reply:** This has been corrected in **Figure 8**.